# Quantifying the Dialect Gap and its Correlates Across Languages

**Anjali Kantharuban, Ivan Vulić, and Anna Korhonen**
Language Technology Lab
University of Cambridge
{atk30,iv250,alk23}@cam.ac.uk

## Abstract

Historically, researchers and consumers have noticed a decrease in quality when applying NLP tools to minority variants of languages (i.e. Puerto Rican Spanish or Swiss German), but studies exploring this have been limited to a select few languages. Additionally, past studies have mainly been conducted in a monolingual context, so cross-linguistic trends have not been identified and tied to external factors. In this work, we conduct a comprehensive evaluation of the most influential, state-of-the-art large language models (LLMs) across two high-use applications, machine translation and automatic speech recognition, to assess their functionality on the regional dialects of several high- and low-resource languages. Additionally, we analyze how the *regional dialect gap* is correlated with economic, social, and linguistic factors. The impact of training data, including related factors like dataset size and its construction procedure, is shown to be significant but not consistent across models or languages, meaning a one-size-fits-all approach cannot be taken in solving the dialect gap. This work will lay the foundation for furthering the field of dialectal NLP by laying out evident disparities and identifying possible pathways for addressing them through mindful data collection.

## 1 Introduction

Across the globe, humans speak over seven thousand unique languages (Eberhard et al., 2022). Many of these languages contain a plethora of internal variation due to the environmental, cultural, and socioeconomic diversity inherent to large populations (Honkola et al., 2018). These dialects are categorised into two groups: standard and non-standard (Trudgill, 2004). Standard dialects are, by definition, supported by governmental and educational institutions resulting in more opportunities of all kinds for their speakers. On the other hand, speakers of non-standard and minority dialects find themselves at a disadvantage compared to their counterparts (Trudgill, 1979). These effects are compounding; those who speak minority dialects are provided fewer opportunities to advance socially and economically, resulting in a self-fulfilling cycle of oppression. Many people who speak a minority dialect as a first language find themselves modifying their use of language throughout their life to appear to belong to the group of standard dialect speakers, much to the detriment of the maintenance of dialectal diversity (Carlson and McHenry, 2006). In losing these dialects, we lose not only the form of expression itself but aspects of the unique culture and society it belongs to (Fishman, 2007).

NLP has been moving in recent years to provide more methods of communication, both between people and with digital systems. In doing so, it has been bridging information- and access-based gaps for people in many historically marginalized communities (Bouillon et al., 2021; Mariani et al., 2022; Zhang et al., 2022). However, it is important to acknowledge that variation within languages is rarely addressed in mainstream tools. Modern systems that do provide access to variants still focus on wealthy, standard dialects, such as British, Australian and American English, while disregarding commonly spoken minority dialects like African American English. Speakers of under-resourced dialects, variants of both high- and low-resource languages with little available training data, face language barriers when using many of the tools taken for granted by speakers of well-resourced dialects. This reduced accessibility further entrenches existing disparities by continuing the historical trend of disenfranchising speakers of minority dialects (Trudgill, 1979).

In this paper, we examine the performance of large language models (LLMs) from two crucial multilingual tasks, machine translation and automatic speech recognition, across a diverse set of dialects and analyze the linguistic, socioeconomic, and computational factors that may contribute to

the dialect gap. This study determines that the largest indicator for better performance for under-resourced dialects is linguistic proximity to well-resourced dialects, regardless of the size or wealth of the dialects' speaker base. This connection is predicted to be due to the lack of dialectal data included in training large language models, leading to dialects performing better or worse on the basis of incidental similarity to the dialect used in training. Unfortunately, the size of the performance gap and the amount/makeup of data required to overcome it is not predictable from external information about the language since it varies across task, model, and environment. As a result, further analysis will need to be done by researchers for individual systems to examine how the dialect gap can be closed for their work through a unique combination of higher-quality, larger, and more balanced datasets.

## 2 Dialect Diversity in Research

**Studies in Linguistic Diversity** A significant problem in the study of linguistic diversity across NLP is the lack of attention paid to language variation. In the past few years, increased awareness has been drawn within the NLP community to the disparities present in modern research. In particular, researchers have begun to notice the relative lack of papers that address languages spoken outside of Europe and East Asia, even in subfields like multilingual NLP (Blasi et al., 2022; Joshi et al., 2020; Ruder et al., 2022; Søgaard, 2022).

While these works offer insight into the disadvantages faced by speakers of under-resourced languages, they still are discussed under the assumption that if languages were appropriately attended to, all their speakers would gain equal access to NLP tools. Similarly, they present their comparisons as if all speakers of well-resourced languages, especially English, have superior access to tools. Unfortunately, this is not necessarily the case. Two-thirds of English's one-and-a-half billion speakers are second-language (L2) speakers (Eberhard et al., 2022). Many L2 speakers struggle with NLP systems due to their accent or their use of code-switched and mixed language. Even many first-language (L1) speakers, such as speakers of African American or Scottish English, do not see their native dialect supported by speech, dialogue, or translation systems and are forced to mask their natural speech patterns, which is harmful to their mental health and sense of identity (Johnson et al.,

2022; Santiago et al., 2021). As such, existing evaluations of linguistic diversity in NLP are fundamentally incomplete.

**Dialectal Models** The advent of large language models has made it possible to train models that perform well on even low-resource languages (Aharoni et al., 2019; Conneau et al., 2020). The term LLM is not strictly defined, but in this study, we use it to refer to multilingual Transformer-based systems pretrained on large amounts of scraped internet data and finetuned for specific tasks. In these systems, under-resourced languages have their training supplemented by this unannotated, scraped data and cross-lingual transfer (Dabre et al., 2020). The performance gain seen by low-resource languages when using LLMs does not extend to under-resourced variants of languages.

Some LLMs provide allocational support for dialects by treating them as separate languages but their performance is not necessarily comparable to that of the standard form. As an example, Arabic speakers often write in their native dialects when communicating casually online, a phenomenon noted by both the linguistic and NLP research communities (Alshutayri, 2017; Abdul-Mageed et al., 2018). Still, attempts by social media to translate Arabic posts are far less successful than their attempts on French and English, despite many consumer translation systems offering support for major regional dialects of Arabic (Harrat et al., 2019). For dialects outside of those explicitly included in systems, this problem is only exacerbated by a lack of allocational support.

**The Data Problem** The same marginalised languages that face lower performance at the hands of LLMs also face a larger data problem across dialects. Most of the task-annotated data available online for low-resource languages comes from religious texts, government documents, or multinational newspapers (Agić and Vulić, 2019; Skadiņš et al., 2014; Chen et al., 2020). These sources often use a formal manner and avoid dialectal markers, especially when their target population is mostly diglossic and has already had to learn a more standard dialect for survival in the modern world (Alshutayri, 2017; Abdul-Mageed et al., 2018). As a result, the LLMs trained on this data are not built to function on minority dialects and have unclear performance capabilities. Before this problem can be solved, questions must be answered about the amount, quality, and type of data needed to over-

| Task | Models | Languages | Metrics |
|------|--------|-----------|---------|
| **Machine Translation (MT)** | Google NMT (Johnson et al., 2019) Meta NLLB (Costa-jussà et al., 2022) Helsinki OpusMT (Tiedemann and Thottingal, 2020) | Arabic (16), Finnish (2), Mandarin (2), German (3), Malay (3), Portuguese (2), Swahili (2) | BLEU, SentenceBERT Similarity |
| **Automatic Speech Recognition (ASR)** | Google USM (Zhang et al., 2023) Google STT (Chiu et al., 2018) OpenAI Whisper (Radford et al., 2022) Meta XLS-R (Conneau et al., 2020) | Arabic (8), Spanish (8), Bengali (3), Georgian (2), Tamil (5), Telugu (4), Tagalog (3) | WER, CER |

Table 1: Tasks addressed in this study along with models, languages (with the number of dialects), and metrics.

come the data problem. The survey done in this paper across languages provides insight into how well dialects perform 'as is' and identifies that linguistic and socioeconomic knowledge should be leveraged to inform future decisions on data collection and usage.

## 3 Tasks

The two tasks evaluated in this paper are machine translation (MT) and automatic speech recognition (ASR). These tasks are some of the few with sufficient data for evaluation of dialects and both have a focus on increasing access to people, tools, and information by removing linguistic barriers (Jin et al., 2021). They are also safe tasks to use as a starting point because they do not deal with personal information or abusive language. The list of models, languages, and metrics used in the evaluation of each task can be found in Table 1. More information about the datasets and languages used can be found in Appendix A. In total, there are six model versions evaluated for each task and 30 dialects across 7 languages compared for MT and 33 dialects across 7 languages compared for automatic speech recognition. Other than Tamil and Telugu for ASR, each language is taken from a different language family in order to extract information that is independent of specific linguistic features.

### 3.1 Machine Translation

Machine translation is already used in domains such as medicine, law, and information as a method of increasing access to systems (Büttner et al., 2022; Vieira et al., 2021). A leader in the field of multilingual MT is Meta's No Language Left Behind (NLLB), a model that claims "safe, high-quality results" for two hundred languages (Costa-jussà et al., 2022). The specific version of the model evaluated in this study is the distilled 600M

parameter variant[1].

Another popular MT model is Google's Neural Machine Translation (NMT), which is available for use through Google Cloud API (Johnson et al., 2019). NMT is a widespread consumer tool, to the point that Google has had to parse out bitext generated using it when scraping internet data for training (Ni et al., 2022).

We also evaluate the University of Helsinki's OpusMT, a model based on MarianMT and trained on Wikimedia monolingual and multilingual text (Tiedemann and Thottingal, 2020). This model is an interesting comparison to NLLB and NMT because it is not an LLM and represents a different approach - covering more languages at the cost of performance across the board. This model was constructed in an academic setting with a more transparent set of training data and significantly fewer parameters. All evaluations are conducted with English as either the target or source language due to data constraints.

Evaluation metrics are a biased measure of output quality and fluency but are required to empirically showcase the dialect gap. To reduce some of the negatives associated with each metric, we report two types of metrics that measure different aspects of the output. The first metric is a BLEU score, which is a classic n-gram evaluation technique for translation (Papineni et al., 2002). Secondly, a representation-backed metric is used to determine the semantic similarity between two sentences since MT is a task with multiple possible solutions. Most semantic similarity metrics are based on transformer embedding models, so we use a multilingual variant of SentenceBERT[2] (Reimers and Gurevych, 2019). Full results for both metrics are reported in Appendix C.

---

[1] huggingface.co/facebook/nllb-200-distilled-600M

[2] huggingface.co/sentence-transformers/paraphrase-multilingual-MiniLM-L12-v2

## 3.2 Automatic Speech Recognition

Automatic speech recognition (ASR) is a task that is important in bringing access to those who are unable or disinclined to communicate through text (Ahn and Lee, 2016; Doumbouya et al., 2021). As of late, representation learning and LLMs for end-to-end ASR have been becoming more common. Many models are trained on unsupervised audio data and then finetuned for specific tasks. This is the case for Meta's XLS-R, a model that is trained on thousands of hours of speech data across languages (Conneau et al., 2020). We evaluate both on a multilingual variant[3] and a monolingual variant of the 300M parameter base model[4] finetuned on a single language at a time using the Common Voice dataset (Ardila et al., 2020).

Another model examined is OpenAI's Whisper, which is trained on a combination of existing ASR datasets and automatically generated transcripts scraped from the internet (Radford et al., 2022). The version of the model tested here is the medium variant[5]. Like XLS-R, the Common Voice dataset was used to finetune this model by language for an additional evaluation (Ardila et al., 2020).

Lastly, Google has released two ASR models: the monolingual Speech-To-Text (STT) and their newer multilingual Universal Speech Model (USM) (Chiu et al., 2018; Zhang et al., 2023). These models were both evaluated through Google Cloud API because neither has been released for open-source use. STT in particular functions as a good comparison to the LLMs evaluated here because it is an older, monolingual model. Overall, six models will be compared - three "monolingual" models (including those finetuned for a specific language) and three multilingual models.

While there has been discussion on whether word error rate (WER) and character error rate (CER) adequately predict performance, no better system has been used by the community at large (Favre et al., 2013). There have been other options, but these are primarily for downstream end-to-end tasks, such as speech translation, natural language understanding, and information retrieval (Kim et al., 2021; Roy, 2021). For this work, we will stick with the community standard and use WER, with CER scores reported in Appendix C.

---

[3] https://huggingface.co/voidful/
wav2vec2-xlsr-multilingual-56
[4] https://huggingface.co/facebook/
wav2vec2-xls-r-300m
[5] https://huggingface.co/openai/whisper-medium

## 4 Linguistic Analysis of Dialects

There are many ways to identify and quantify the similarity between two variants of a language. Many have been explored in NLP for cross-lingual transfer using features from syntax, lexicon, and morphology (Philippy et al., 2023; Eronen et al., 2023; Lin et al., 2019; Ponti et al., 2019). There have also been studies on dialects in computational linguistics, examining whether dialects are consistent across corpora and registers (Dunn, 2021). A similar method is used in this paper to examine lexical similarity, using Spearman's Rank Correlation Coefficient. This has been used previously to calculate corpus similarity and homogeneity (Kilgarriff and Rose, 1998). In Appendix Figure 3a, the similarity between each dialect and the best-performing variant of that language is shown, as well as the lexical similarities between scripted and conversational samples from each dialect of the Babel dataset.

Additionally, we examine the phonetic similarity of selected ASR datasets, specifically for Arabic and Spanish. Here, random samples were manually annotated for vowel positioning through formant analysis and plotted in the Appendix; see Figure 3b. Then, the average Euclidean distance across vowels between each dialect and the standard form was taken to serve as a measure of phonetic similarity. More details on the exact methodology can be found in Appendix B.

## 5 Dialect-Wise Performance Gaps

Examining the performance across dialects in Figure 1, some trends appear immediately. As mentioned in Appendix A, the dialects evaluated were largely dictated by data availability. As a result, Arabic and Spanish are heavily represented while other lower resource (dialect-wise) languages see coverage of only two to three dialects. This is something that may be reflected as well in the training data for pre-trained models, resulting in Arabic and Spanish both having relatively more even dialectal performance than the other surveyed languages.

For MT, there are steeper performance gaps when translating into the dialect. This makes sense if input robustness is taken into account; in other words, models may be able to handle some level of dialect variation in their input but cannot know to output the non-dominant dialect. Additionally, models that perform better on the standard dialect show steeper drop offs in performance, something

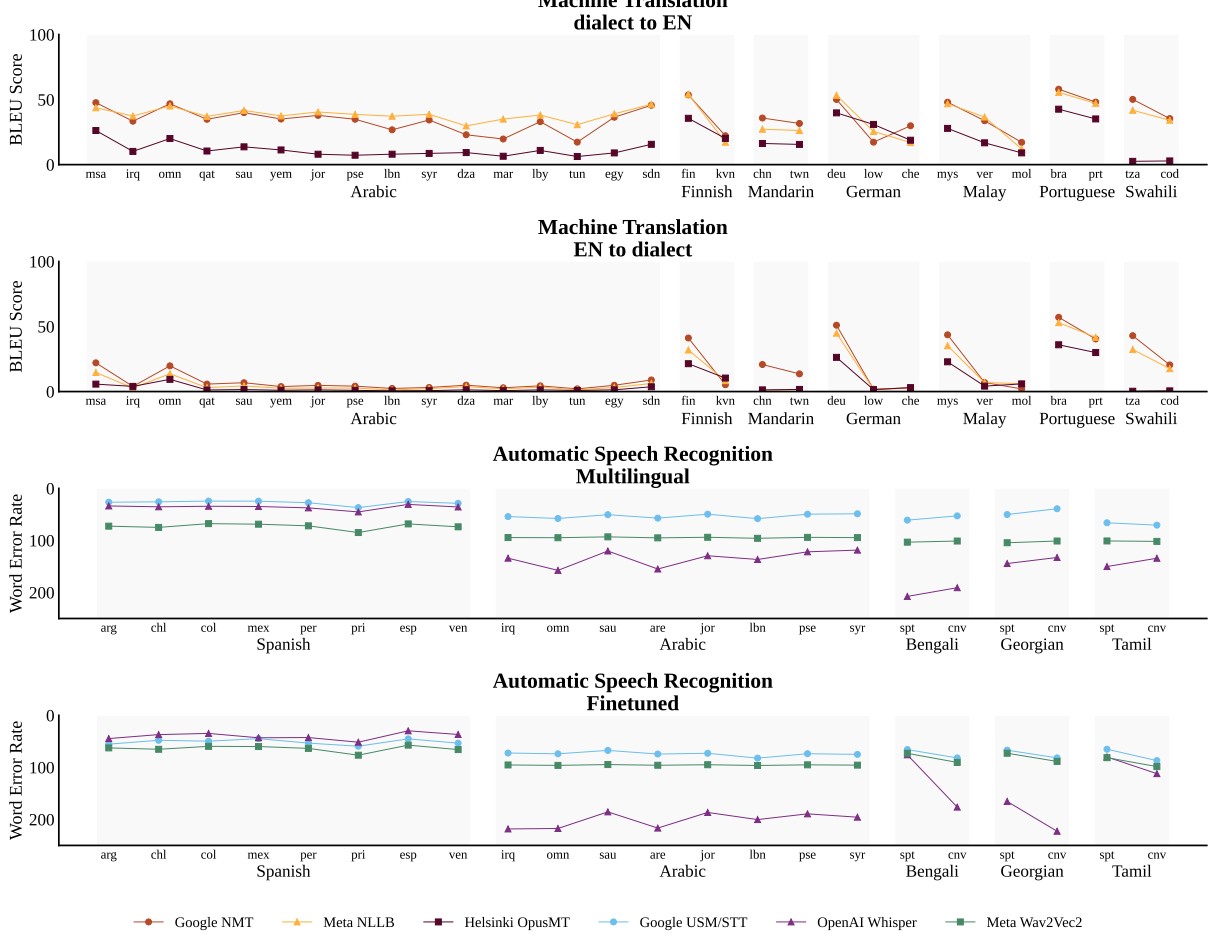

Figure 1: The performance of various dialects across Machine Translation and Automatic Speech Recognition. Within each language the same dataset is used. Because Bengali, Georgian, and Tamil are heavily diglossic, the standard written form (spt) and the best performing spoken form (cnv) are compared rather than regional dialects.

very clearly exemplified across the Finnish dialects. This demonstrates the interesting point that higher performing models - which have access to more parameters and data during training - have greater inequalities in their coverage.

The same trends are not apparent for ASR, where the worst performing model (OpenAI's Whisper) has the highest amount of variance across dialects. Interestingly, all three multilingual models seem to prefer the spoken dialect (cnv), likely due to the fact that they are mostly trained on unsupervised internet data from websites like Youtube. On the other hand, the finetuned models prefer the written dialect (spt), which is understandable since most are finetuned using CommonVoice, a heavily scripted data source.

## 6  Correlations with Proximity to Power

Even within the same task and model, different dialects have different performance disparities, as seen in Figure 1. In order to examine this phenomenon in an equivalent environment, we compare performance across MT using BLEU %, which is the percentage of the best-performing dialect's BLEU score achieved by the minority dialect. Likewise, for ASR, the relative percentage loss of performance for each dialect compared to the standard dialect is used. Note that this means that in Figure 2, a *positive* MT correlation and a *negative* ASR correlation both mean there is positive correlation between the metric and performance.

In choosing metrics for comparison, we aimed to cover the range of economic, social, and linguistic factors that capture the idea of proximity to power. As proxies for wealth, we examine gross domestic product (GDP) for cumulative economic power and GDP per capita for individual economic power (The World Bank, 2023). Socially, we are interested in both population size and how well-

| Metric | Machine Translation | | Speech Recognition | |
|---|---|---|---|---|
| | EN → di | di → EN | Multilingual | Finetuned |
| **Gross Domestic Product** | 0.11 | 0.16 | -0.25 | -0.13 |
| **Gross Domestic Product per Capita** | 0.16 | 0.31* | 0.44* | 0.16 |
| **Population Size** | 0.04 | -0.13 | -0.30 | -0.00 |
| **Human Development Index** | -0.06 | -0.03 | 0.23 | 0.09 |
| **Lexical Similarity** | 0.48* | 0.69* | -0.70* | -0.57* |
| **Phonetic Similarity** | - | - | -0.49* | -0.63* |

Table 2: Pearson correlation coefficients for each language metric. For MT, correlation is calculated against percentage drop in BLEU performance while for ASR, correlation is calculated against percentage increase in WER. As such, the correlations are reversed for WER. Correlations with $p < 0.05$ are marked.

served the population is in education, healthcare, and standard of living, estimated via the Human Development Index (HDI) (US Census Bureau, 2017; United Nations Development Programme, 2021). Lastly, for linguistic factors, we utilize the lexical similarity and phonetic similarity extracted from evaluation data and normalized to a scale from $-1$ (lowest similarity) to 1 (highest similarity). Unfortunately, some economic and social metrics are only reported at the national level, so there is no data for minority dialect groups within countries. As a result, certain dialects (e.g. Kven Finnish, Vernacular Malay) are not included.

In the past, population factors have been shown to loosely correlate with factors such as performance and appearance in NLP research (Blasi et al., 2022). Here, in Figure 2, we see that these correlations do not necessarily hold for dialects. In fact, these results are contradictory to common expectations and narratives, which assume that wealthier, larger, and more educated populations are better served across the board.

**Gross Domestic Product** GDP represents the overall wealth of a speaker population and their economic power in the world. As such, we would expect groups with high cumulative wealth to be well-served by technology. While GDP has a small impact, it varies heavily by model and can't be used as a consistent predictor of performance. Certain models show a relatively consistent positive correlation, such as OpusMT and USM/STT, but others show no correlation at all. Ohers showcase a correlation only in one set of models, such as NLLB which is uncorrelated when translating into English but positively correlated when translating into the dialect. On average, worse-performing models and environments show a stronger correlation, with translation into the dialect being much more correlated than translation into English.

**Gross Domestic Product Per Capita** GDP per capita is an important metric as a proxy for estimating the wealth of individuals in a population and we would expect those with access to wealth to be well-served even if their population is smaller. Surprisingly, it seems to have no impact at all on MT across models, so wealthier minority populations are not better served than poorer ones despite having access to increased resources. In ASR, the result is even more unexpected with wealth correlating negatively with performance.

**Population Size** Population size intuitively would correlate with better performance, but previous studies on language diversity in NLP have shown that even languages with extremely high populations are not well-served if they are impacted by other factors like geographic distance from research institutions and low wealth (Blasi et al., 2022). Here, population size has little impact on MT performance, to the point that certain models show a negative correlation between the two. On the other hand, in ASR there is a strong positive correlation across all models except for the finetuned version of Whisper. This is an unexpected result because Whisper originally showcases a matching positive correlation and is finetuned on the same Common Voice datasets as XLS-R but demonstrates a complete trend reversal. This difference between MT and ASR may be a result of the type of data used for training each and the sources it came from, but further analysis is needed to confirm this.

**Human Development Index** HDI is a measure of how well a population is served in other access-based metrics, such as education, healthcare, and standard of living. It would logically follow that a high HDI would then correlate with better performance, but this does not hold for MT. Instead, MT performance shows no correlation at all with HDI.

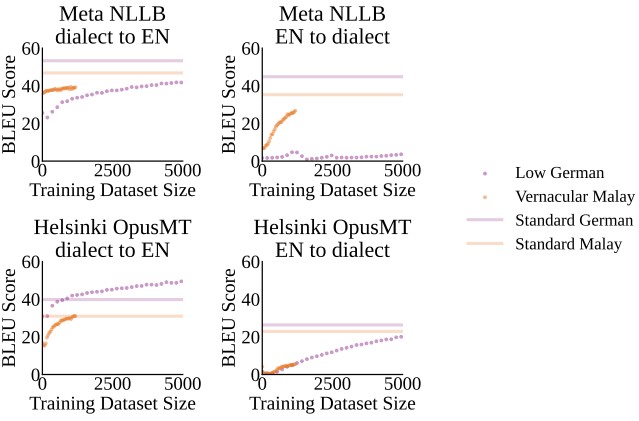

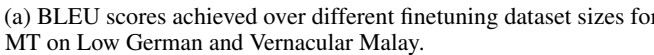

(a) BLEU scores achieved over different finetuning dataset sizes for MT on Low German and Vernacular Malay.

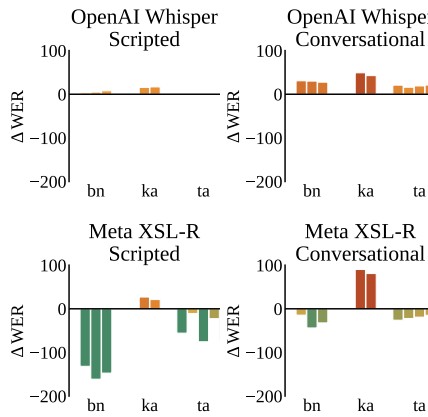

(b) Change in WER after finetuning on scripted Bengali, Georgian, and Tamil data.

Figure 2: Impact of training dataset modifications on performance.

Surprisingly, HDI correlates negatively with ASR performance, so better-educated and healthy minority dialect speakers have a harder time accessing ASR systems despite being otherwise well-served economically and socially.

**Lexical Similarity** Lexical similarity, on the other hand, is very correlated with performance for both MT and ASR. Since dialectal data is not used for training regardless of population features, performance is likely mostly based on linguistic proximity to the standard form. This result is also more robust than the other correlations mentioned here because every dialect of every language evaluated was included since the similarity score was not dependent on external data availability. Again, we also see in MT that the worse-performing directionality (EN → dialect) has a stronger correlation. This is expected in context since these models do not provide allocational support to these dialects, so they are translating into the standard dialect regardless of user intent but they may be robust to some amount of the lexical variation in the input.

**Phonetic Similarity** The importance of linguistic similarity extends to phonetic similarity for ASR, which is strongly positively correlated with performance. Again we see that finetuning on the smaller, scripted Common Voice datasets makes the correlation stronger for XLS-R and Whisper, which suggests that models overfit to the dialects present in training data. It is important to remember that phonetics is a broad area of study in linguistics that encompasses many measures of acoustic similarity, so other forms of analysis may capture even higher impact forms of variation between dialects.

However, these results clearly already show that phonetic similarity plays a large part in determining the performance of dialects.

The results surrounding similarity suggest that the most useful method of addressing the dialect gap may lie in focusing on how to reduce the linguistic distance between the language used at evaluation versus training. In other words, this can be compared to a domain shift problem rather than a multilingual problem. A way to begin is by increasing the dialectal diversity of the training data to cover a larger variety of language patterns.

## 7 The Impact of Datasets

### 7.1 Machine Translation & Dataset Size

For many languages, lower performance in MT is seen in parallel with a smaller dialect gap. As an example, the Mandarin dialects perform comparably on NLLB and OpusMT but the disparity becomes statistically significant under NMT, a model where Mandarin as a whole performs better. This trend suggests that the benefits of larger models and more training data are not equally felt by all dialects due to disparities in the training pipeline — more training data does not solve the dialect gap, *it makes it worse*. The question can then be raised: would training on more specifically dialectal data be sufficient to overcome these disparities?

To answer this question, two languages with enough dialectal data were chosen to finetune NLLB and OpusMT. Each model was trained with thirty different dataset sizes, on three different data subsets per size and three seeds per data subset to ensure that the results were statistically significant.

In Figure 2a, the training curves for each model and translation direction can be seen.

As more data is added, the languages begin to perform better, but not all at the same rate. For example, Vernacular Malay sees relatively little improvement as data is added to train NLLB, but OpusMT's initial training curve is steep. Therefore the same amount of data causes two different outcomes depending on the model architecture and the data it was previously trained on. In some cases, the improvements are marginal, while in others, even a small amount of data is enough to completely overcome the dialect gap. Likewise, the same inconsistencies can be seen between the two directionalities of the same model. Despite both versions of OpusMT being trained on the same Low German data, translation into English sees benefits while translation into Low German remains poor. This makes it clear that the amount of data that dissipates the dialect gap in one situation may not be enough for another model or language.

### 7.2 Speech Recognition & Dataset Makeup

Besides finetuning on more dialectal data, another possible method of addressing the dialect gap is modifying the makeup of training or finetuning data. Across the board, the data used to finetune LLMs for speech recognition is heavily influential on performance. This difference can be seen when comparing the performance of these ASR systems on conversational and scripted samples from the IARPA Babel dataset (Bengali, Georgian, Tagalog, Tamil, & Telugu). The models evaluated here are largely trained on unsupervised speech data from the internet, which mostly comes from unscripted conversational recordings. As a result, the multilingual models perform slightly better on conversational speech. To test the impact of data makeup, XLS-R and Whisper were finetuned for three languages (Bengali, Georgian, & Tamil) on Common Voice, an entirely scripted dataset (Ardila et al., 2020). These languages are all spoken by a diglossic population that uses both a regional dialect and a more linguistically conservative standard written form. As a result, the lexical distance between conversational and scripted samples is farther than might otherwise be expected. In Figure 2b, finetuning on scripted data almost exclusively benefits performance for scripted samples over conversational samples. In some cases, such as with Whisper, this comes at the detriment of performance on conver-

sational samples. This ties back into the impact of lexical variation discussed in Section 6 since both scripted and conversational samples were collected by speakers of the same dialect with similar accents. The low lexical similarity between these dialects amplifies the fact that ensuring the training dataset accurately and fully represents the lexical variations across a language and its dialects is an important step in creating systems that perform well across dialects, domains, and registers.

## 8 Implications of the Dialect Gap

The existence of a dialect gap means that not all speakers are inherently well-served by a tool just because their language is supported. Past analyses examined inequities from the perspectives of multilingualism and therefore likely overestimated the number of speakers benefiting from the current system. As the field moves forward, it is important to step back and remember that languages are not static or monolithic.

Additionally, as we saw, the dialect gap is not identical in severity or structure across every system. This implies that researchers cannot take a one-size-fits-all approach towards solving the dialect gap. This issue needs addressing in different ways depending on the task and the existing state of the gap. A large component of dialect gaps is based on datasets — both dataset size and dataset makeup. As the NLP community moves towards furthering research for medium- and low-resource languages, discussions must be had on both collecting sufficient amounts of dialectal data and capturing the natural variations of every language by ensuring that data is collected from diverse populations. Appreciating and accounting for variation not only makes our systems more robust but supports groups that face marginalization in other ways.

## 9 Conclusion

This work examined an important subspace in NLP by evaluating the (regional) dialect gap present in tasks with the highest likelihood of impacting speakers directly. Still, there are countless LLMs which have been rapidly gaining popularity in the past few years with the release of open-ended dialogue and image models. Most tasks outside of MT and ASR do not have the data necessary to analyze the impact of language variation but as more data is collected and annotated, this may change. As a direct continuation of the line of inquiry started

in this work, multi-dialectal analyses of the dialect gap across a wider variety of tasks should be next.

For MT and ASR, the next steps are two-fold. Firstly, the datasets used for evaluation and finetuning in this work were primarily determined by availability, but using a broader and higher-quality set of samples may lead to the rise of other interesting trends. Additionally, to address the dialect gap identified here, there is a clear path forward that involves collecting more dialectal data and ensuring it is representative of the languages and dialects it aims to serve. This should be done in conjunction with speakers of the language, linguists, and members of the NLP community to maximise utility while minimising the burden or harm on the speaker population. Lastly, this analysis is hardly complete. As new LLMs come out, it is on the developers of these tools and the researchers behind them to continuously produce evaluations around language diversity to ensure that the benefits these LLMs bring do not come at the cost of access for minority dialect speakers.

## Limitations

**Dataset Size & Quality Factors** Dataset size is a very significant factor when evaluating models and drawing language-wide conclusions. While the languages seen in this work had enough data for evaluation, very few provided enough data for finetuning LLMs and none provided enough to train a model from scratch. As a result, models were largely evaluated out of the box, which serves to identify performance gaps as they may appear in non-academic use cases but does not fully address solutions to this problem.

Likewise, dataset quality makes a massive impact on the result of training and evaluation. Because the number of available datasets was already quite low, crowd-sourced datasets such as Tatoeba were used without additional filtering, which may result in increased noise due to improper annotations. For some datasets, such as the IARPA Babel speech dataset, there was filtering done but spontaneous speech data in general is often paired with background noise and distortion, causing a further drop in performance.

Some languages have several datasets available, but because these datasets were not all collected with the same methodology (and therefore similar errors and distortions), they were not directly comparable so only one dataset was used or the language was not evaluated. Spanish speech, for example, has been recorded in the OpenSLR, CALLHOME, and Fisher datasets but CALLHOME was chosen alone to be used. On the other hand, a multitude of English accent and dialect datasets are available for speech, but because each was collected independently, they again could not be directly compared and were therefore omitted. Lastly, some languages supported by models (Telugu and Tagalog) were not present in the Common Voice finetuning dataset used for the ASR experiments and were therefore omitted from a large part of the discussion surrounding dataset makeup.

**Computational Restraints** Many of the models evaluated are large industry models, with hundreds of millions if not billions of parameters. Naturally, as an academic institution, we were limited in the computational power made available to train these models; certain models were so large that even with a batch size of one they are incapable of running on the machines we have available. If we had greater computational power available, we would have run our evaluations on the largest version of each model to provide a picture of the most state-of-the-art performance for each task and finetune these larger models longer. On the other hand, many minority dialect speakers do not have the economic resources to train or finetune super-massive models, so the evaluation of more accessible models is an appropriate reflection of what is available to these speakers. In the future, with access to greater resources, the evaluation of more systems and larger models, along with the evaluation on other user-facing tasks (Ruder et al., 2023), again through the optics of regional dialects, ould be a valuable extension of this work.

## Ethics Statement

Dialectal NLP research is a burgeoning field without many precedents set for ethical research, but direction can be taken from the field of multilingual NLP for how to work with the languages of minoritised groups ethically. In this paper, the issue of ethics was largely sidestepped through the use of anonymised, public, and voluntarily collected datasets and the evaluation of tasks with a low likelihood of causing harm. Additionally, despite the importance of idiolects and moving beyond regional dialects, we purposefully did not work with dialects connected to identity features that may put people at risk, such as sexuality, gender, and reli-

gion. Even as this paper supports the collection of larger and more representative datasets, these arguments do not apply in cases where it would be against the wishes or best interests of the groups involved.

## Acknowledgements

This work has been in part supported by the UK Research and Innovation (UKRI) Frontier Research Grant EP/Y031350/1 (the UK government's funding guarantee for ERC Advanced Grants) awarded to Anna Korhonen at the University of Cambridge. The work of Ivan Vulić has been supported by a personal Royal Society University Research Fellowship *'Inclusive and Sustainable Language Technology for a Truly Multilingual World'* (no 221137; 2022-). The work of Anjali Kantharuban has been supported by a Gates Cambridge Scholarship.

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

## A Languages

In total, there are seven languages evaluated in this study per task and over thirty dialects. The details of these datasets are discussed in Section A, but for easy reference, a table listing the datasets, the languages included, and any dialects is provided in Table 3. Additionally, any abbreviations used in figures throughout the paper are included, with language abbreviations pulled from ISO 639-1 and dialect abbreviations either based on the ISO 3166 country codes or created based on the regional names provided in the dataset.

**AraBench** Arabench is a dataset collected by the Qatar Research Computing Institute to encourage research into machine translation for Arabic dialects (Sajjad et al., 2020). The dataset includes parallel text data grouped by region, nation, and city from religious, media, and speech sources. Across the board for Arabic, Modern Standard Arabic (MSA) outperforms the vast majority of dialectal forms, even though social media's rise has led to dialectal forms of Arabic being used in written communication more often. In fact, in personal chat, dialectal forms of Arabic are represented more than MSA (Chelghoum, 2017).

| Dataset | Language | ISO 639-1 | Dialects (Abrv.) |
|---|---|---|---|
| **AraBench** | Arabic | ar | Modern Standard (msa), Iraqi (iq), Omani (om), Qatari (qa), Saudi (sa), Yemeni (ye), Jordanian (jo), Palestinian (ps), Lebanese (lb), Syrian (sy), Algerian (dz), Moroccan (ma), Libyan (ly), Tunisian (tn), Egyptian (eg), Sudanese (sd) |
| **Region-Aware MT** | Mandarin | zh | Mainland (cn), Taiwanese (tw) |
| | Portuguese | pt | Brazilian (br), European (pt) |
| **Tatoeba** | Finnish | fi | Standard (fi), Kven (kv) |
| | German | de | Standard (de), Low (lo), Swiss (sw) |
| | Malay | ms | Standard (ms), Vernacular (vn), Moluccan (mo) |
| | Swahili | sw | Coastal (co), Congolese (cg) |
| **Conversational Telephone Speech** | Arabic | ar | Iraqi (iq), Omani (om), Saudi (sa), Emirati (ae), Jordanian (jo), Lebanese (lb), Palestinian (ps), Syrian (sy) |
| **CALLHOME** | Spanish | es | Argentinian (ar), Chilean (cl), Colombian (co), European (es), Mexican (mx), Peruvian (pe), Puerto Rican (pr), Venezuelan (ve) |
| **Babel** | Bengali | bn | Kamrupa (kp), Radha (rd), Varendra (vd) |
| | Georgian | ka | Eastern (en), Western (wn) |
| | Tagalog | tl | Central (cn), Northern (nn), Southern (sn) |
| | Tamil | ta | Central (cn), Madurai (md), Northern (nn), Southern (sn), Western (wn) |
| | Telugu | te | Central (cn), Northern (nn), Southern (sn), Western (wn) |

Table 3: The datasets, languages, dialects and abbreviations used throughout this paper.

**Google Region-Aware MT**   The dataset used for Mandarin and Portuguese MT is Google's Region-Aware Machine Translation Dataset, a small benchmarking dataset for few-shot translation for these two high-resource languages (Riley et al., 2022). The dataset consists of Wikipedia articles that exist in both English and the target dialects.

**Tatoeba**   The Tatoeba datasets are a set of translation datasets crowdsourced by the Tatoeba organisation[6] (Tiedemann, 2020). Languages have varying amounts of data, ranging from over a million sentences in English to fewer than ten for languages such as Sindhi. The majority of parallel text available in these datasets is for low-resource languages paired with English, so most translation systems trained on this data use English as either the source or target language. While Tatoeba is a project focused on language diversity, there are efforts within it to include some major regional dialects. Dialects of languages such as German (Standard, Low, & Swiss), Finnish (Standard & Kven), Malay (Standard, Venacular, & Moluccan), and Swahili (Coastal & Congolese) are available and included in our analysis.

**Conversational Telephone Speech**   The dataset used to evaluate Arabic for ASR is the Conversational Telephone Speech (CTS) dataset, a spontaneous spoken language dataset with transcriptions available through the Linguistic Data Consortium (Appen Pty Ltd, 2006c,d, 2007a,b, 2006a,b). This set of datasets encompasses the Gulf (Emirati, Omani & Saudi), Mesopotamian (Iraqi), and Lev-

antine (Jordanian, Lebanese, Palestinian & Syrian) dialects of Arabic.

**CALLHOME**   The dataset used to evaluate Spanish ASR is the CALLHOME telephone speech corpus, which encompasses several primarily Latin American Spanish datasets (Canavan and Zipperlen, 1996; Wheatley, 1996). In this work, we specifically focus on eight dialects: Argentinian, Chilean, Columbian, European, Mexican, Peruvian, Puerto Rican, and Venezuelan Spanish. Spanish is an interesting case of two "standard" forms, Mexican and European Spanish.

**IARPA Babel**   The IARPA Babel dataset[7] was a large set of speech recognition datasets collected for medium- and low-resource languages to make ASR effective on a broader set of languages. While there are many languages included, most are not currently supported by large ASR systems, so five have been selected for these experiments: Bengali (Kamrupa, Radha, & Varendra), Georgian (Eastern & Western), Tagalog (Central, Northern, & Southern), Tamil (Central, Madurai, Northern, Southern, & Western), and Telugu (Central, Eastern, Northern, & Southern). Each of these languages is written in a different script, increasing the difficulty. It is important to note that the Babel project's goal was not to represent all dialects of these languages, so certain significant dialects (e.g. Sri Lankan and Singaporean Tamil for the Tamil dataset) are omitted due to the focus on a single major region. The Babel dataset divides each language into two to five regional dialects, with half the samples being

[6]https://tatoeba.org/

[7]https://www.iarpa.gov/research-programs/babel

scripted speech and the other half being spontaneous utterances. While we are primarily interested in spontaneous speech for this work, all of these languages have a distinct written form which is considered a separate dialect. As a result, the scripted audios are kept but separated from the spontaneous samples to examine how much lexical variation (between the scripted and spontaneous samples of each dialect) impacts performance in comparison to phonetic variation (across dialects).

## B    Linguistic Analysis

**Lexical Similarity**    The lexical similarity between dialects is calculated by splitting the full dataset for each dialect and randomly sampling it into halves one hundred times. The top hundred most frequently occurring words are then ranked by frequency to generate a vocabulary list. For each dialect pair of interest, the position of each word in the ranking is compared using the following formula:

$$s = 1 - \frac{6 \sum d_i^2}{n(n^2 - 1)}$$

Here, $d_i$ is the difference in ranking between each $i$ word in the list and $n$ is the number of words being compared (100 in this case). If a word appears in one list but not the other, the ranking given is $n + 1$. The same process is utilized for calculating homogeneity, except both halves are taken from the same corpus. Homogeneity is calculated to ensure that there isn't a large degree of internal variation that is influencing the degree of lexical cross-similarity between dialects. While this analysis can be impacted by domain mismatches, by focusing on only the top one hundred words we deal primarily with very common words that are less likely to be specialized vocabulary.

**Phonetic Similarity**    The phonetic similarity between dialects is computed in a time-intensive method for this paper, so the results of this computation should be considered preliminary until further annotations can be collected. For each language of interest, the primary vowels of that language are determined based on the orthographic choices made in that language. This amounted to three vowels in Arabic (/a/, /u/, & /i/) and five vowels in Spanish (/a/, /e/, /i/, /o/, & /u/). For simplicity, other highly sonorant phonemes, such as /j/, are not included, although some may consider them vowel-like. From this, the set of samples from each dialect where all the language's vowels are present

is extracted in order to reduce discrepancy due to variations in distortion or recording environments as much as possible. Ten samples per dialect, from unique speakers (when possible) are then annotated with the first ($f1$) and second ($f2$) formant, which provides information on the height and backness of the vowel. While roundedness can be gleaned from further formants, this adds another level of complexity so it was not considered in this study. The height of the vowel was then set as $f1$ and the backness was set as $f2 - f1$. The average distance between the dialects' mean vowel positions and those of the best-performing dialect was taken to represent phonetic similarity.

## C    Full Evaluation Results

**Machine Translation**    Here, we include the full results achieved across dialects for transparency. The BLEU score results can be seen in Table 4 and the semantic similarity results can be seen in Table 5. The label "di → EN" refers to translation into English from the dialect and "EN → di" refers to translation from English into the dialect. The closest language tag was used when possible, such as regional dialect tags for Arabic in NLLB.

**Automatic Speech Recognition**    Likewise, full results are included for both the raw and finetuned versions of our ASR models. In Table 6, the word error rates (WER) are provided and in Table 7 the character error rates (CER) are provided. When there is a "+ CV" included in the model title, that refers to further finetuning on the Common Voice dataset. Additionally, the "spt" and "cnv" titles refer to the scripted and conversational splits of the languages in the IARPA Babel dataset. Spanish and Arabic are only evaluated with conversational samples, so there is no data in the scripted column.

## D    Correlations

The plot comparing various metrics with dialect performance can be found in Figure 4. The vertical axis for the MT plots is BLEU %, or the percentage of the standard dialect's BLEU score achieved by the minority dialect. Likewise, the vertical axis for the ASR plots is WER %, or the percentage worse WER achieved by the minority dialect, since a lower WER is better. This means that a higher score on the two left columns is better, while a lower score on the two right columns is better. As you can see heuristically, the only obvious corre-

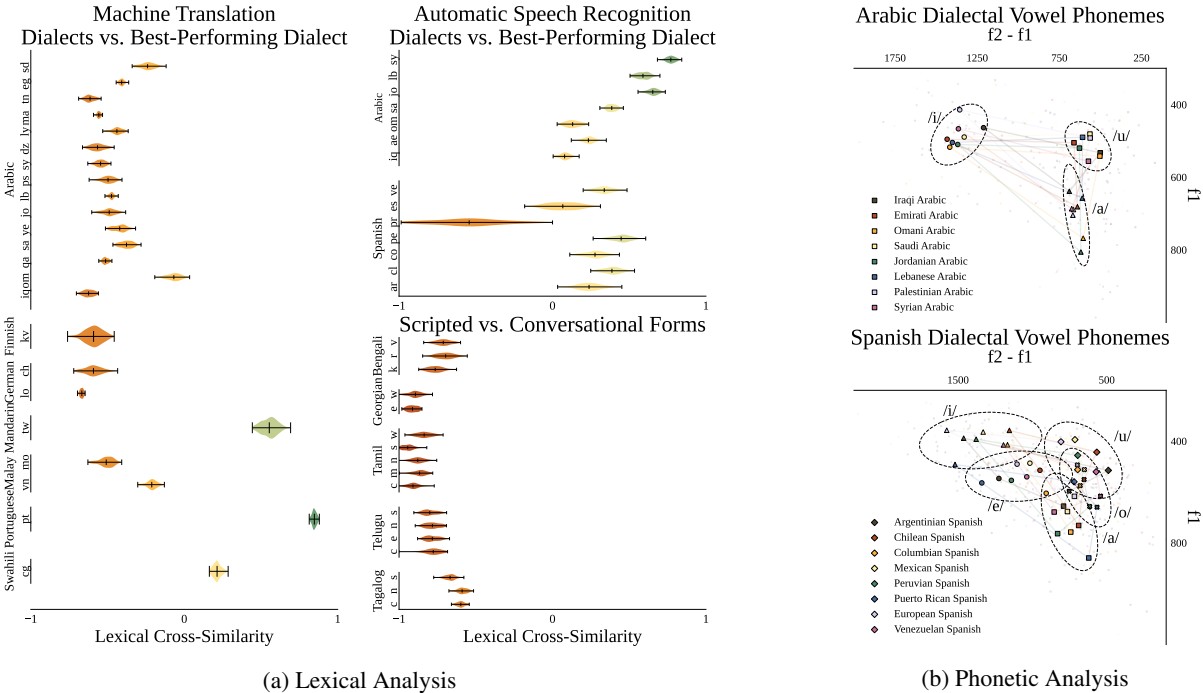

Figure 3: Linguistic similarity measures comparing dialects. Left: Lexical similarity, taken either between each dialect and the best-performing dialect from that family or between spontaneous and conversational samples of the same dialect. Right: Vowel positions for the most common vowels in two languages across dialects.

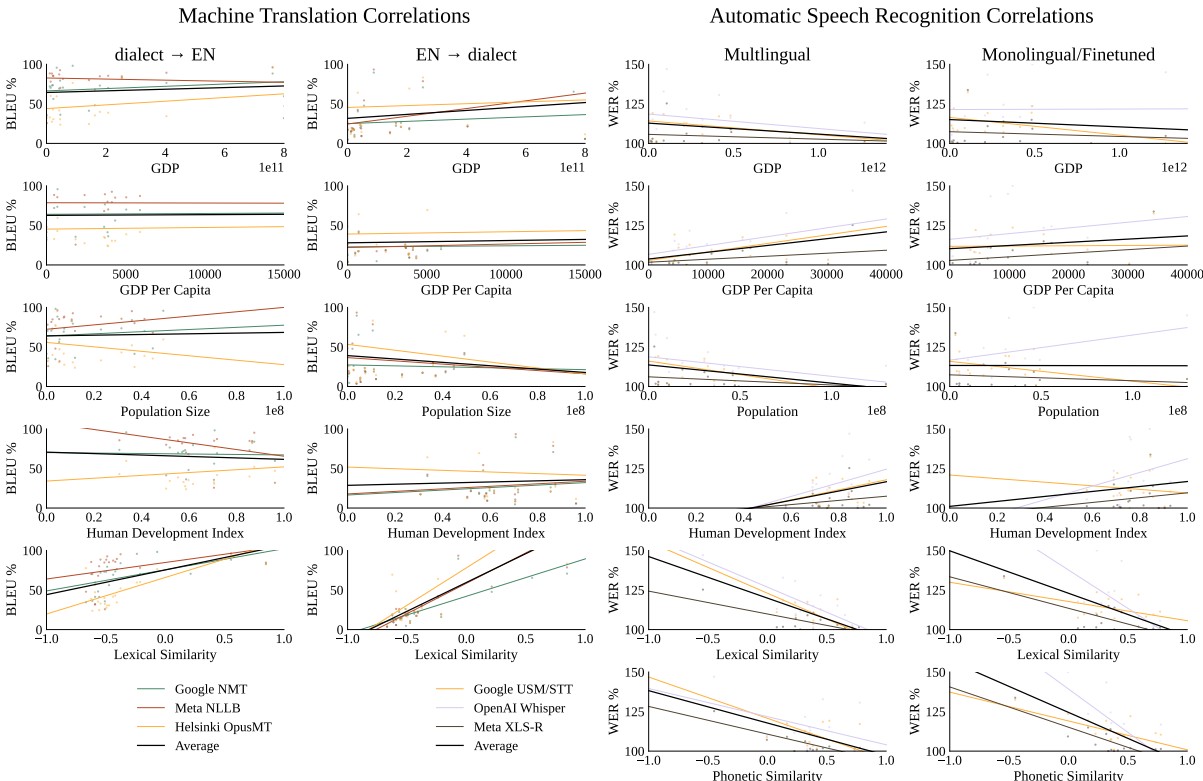

Figure 4: Correlation between various metrics and the relative performance of minority dialects, normalised to be relative to the best performing dialect. Note: The MT graphs are with respect to BLEU scores, where a higher value means better performance, but the ASR graphs are with respect to WER, where the inverse is true. Linguistic similarity is positively correlated with better performance in both systems.

lates are both linguistic measures, which is confirmed in our calculations.

# E    Concerns on Ethics in Dialect Research

Dialectal research is a relatively new field in NLP and as such, it is important to consider the ethical impacts of conducting it. In the past, researchers have examined the ethics of many aspects of NLP as they relate to marginalized and vulnerable populations, including raising concerns about sources of data (Olson et al., 2023; Rogers et al., 2021; Shmueli et al., 2021), evaluations of representational bias (Hutchinson et al., 2020; Lalor et al., 2022; Sun et al., 2019), and bringing awareness to allocational bias (Aji et al., 2022; Blasi et al., 2022). Additionally, there has been significant work done in the medical domain, where the population being served is particularly vulnerable to both coercion and harm from publicized data (Don't Walk et al., 2022; Thompson et al., 2021; Suominen et al., 2007). We can take inspiration from these prior works when examining the ethical pitfalls of dialectal research, since minority dialect communities face many of the same hurdles as other minoritized groups, especially when their language reflects aspects of their identity that face persecution such as gender, race, or class.

Currently, the issues that impact minority dialect speakers are largely around representational and allocational bias. Speakers of certain dialects are viewed as less intelligent or more difficult to understand, leading to systems classifying them as such when it comes to usages such as call screening or academic evaluations (Koenecke et al., 2020; Wassink et al., 2022). Important tools in use, such as automated emergency service or medical phone systems, may not function well on their dialects, reducing access to necessary care. Additionally, these biases are perpetuated externally towards users as well; female voices are often used when building personal assistants, which continues the societal trend of female systems being "bossed around" or acting in a subservient manner. This is especially the case since there has been no strong correlation in increased trust or comfort when personal assistants are of a specific gender, but the vast majority of consumer systems use female voices anyway (Tolmeijer et al., 2021). As dialect research improves, there is unfortunately room for these problems to worsen. Currently, gender is easy to specify when generating synthetic speech, but

in the future if ethnicity or socioeconomic class was equally able to be designated, what patterns would emerge in consumer systems? These tools may result in maintaining stereotypes regarding subservience and the "place" of certain groups in society.

Another aspect to consider is the result of datasets and tools that parse certain identity markers falling into the wrong hands. A key example of this in another domain, facial recognition processing, is the Stanford "Gaydar" which was purported to outperform humans in predicting sexual orientation from facial structure (Wang and Kosinski, 2018). This system was denounced by multiple prominent LGBTQ organizations for both its methodology and the risk it places on closeted members of the community since it could be used to out members living in unfriendly places (Levin, 2017). Likewise, in China, similar facial recognition systems have already been used to track the movement of Uighurs, a Muslim minority group that has been facing prosecution by the government (Mozur, 2019). As such, the existence of tools to *identify* a minority are unfortunately inextricably linked with the ability to *repress* that minority. Currently, there are no highly accurate systems that can classify most personal identity features from text alone, but as we work on the identification and usage of dialect data, this may soon change. When conducting this research, it is important to have a deep understanding of the communities impacted by these tools beyond their academic contribution and weigh the consequences of releasing such tools for public use against the value of open and collaborative science.

| Language | Dialect | Google NMT | | Meta NLLB | | Helsinki OpusMT | |
|---|---|---|---|---|---|---|---|
| | | di → EN | EN → di | di → EN | EN → di | di → EN | EN → di |
| **Arabic** | MSA | 47.64 | 22.07 | 43.71 | 14.64 | 26.22 | 05.72 |
| | Iraqi | 33.38 | 03.94 | 37.37 | 02.82 | 10.16 | 03.97 |
| | Omani | 46.78 | 19.77 | 45.07 | 13.68 | 20.05 | 09.40 |
| | Qatari | 34.84 | 05.75 | 37.08 | 03.03 | 10.46 | 01.23 |
| | Saudi | 39.98 | 6.79 | 41.63 | 04.32 | 13.67 | 01.54 |
| | Yemeni | 35.05 | 03.84 | 37.45 | 02.53 | 11.28 | 01.01 |
| | Jordanian | 37.86 | 04.75 | 40.46 | 02.76 | 08.00 | 01.29 |
| | Palestinian | 34.88 | 04.16 | 38.62 | 02.68 | 07.22 | 00.94 |
| | Lebanese | 26.76 | 02.39 | 37.22 | 01.23 | 08.02 | 00.55 |
| | Syrian | 34.31 | 03.14 | 38.69 | 02.27 | 08.63 | 00.68 |
| | Algerian | 22.95 | 04.90 | 29.76 | 03.79 | 09.27 | 01.38 |
| | Moroccan | 19.69 | 02.91 | 34.99 | 02.08 | 06.45 | 00.71 |
| | Libyan | 33.03 | 04.40 | 38.13 | 03.43 | 10.93 | 01.31 |
| | Tunisian | 17.29 | 02.08 | 30.70 | 01.35 | 06.23 | 00.83 |
| | Egyptian | 36.44 | 04.81 | 39.06 | 03.12 | 09.00 | 01.34 |
| | Sudanese | 45.62 | 08.92 | 46.40 | 06.20 | 15.61 | 03.67 |
| **Finnish** | Standard | 53.58 | 41.18 | 54.10 | 31.95 | 35.61 | 21.40 |
| | Kven | 22.19 | 05.23 | 17.24 | 08.35 | 20.13 | 10.38 |
| **Mandarin** | Mainland | 35.82 | 20.79 | 27.22 | 00.90 | 16.27 | 01.29 |
| | Taiwanese | 31.65 | 13.60 | 26.23 | 01.48 | 15.54 | 01.64 |
| **German** | Standard | 50.10 | 51.01 | 53.28 | 44.88 | 39.78 | 26.30 |
| | Low | 17.28 | 01.78 | 25.53 | 01.66 | 30.90 | 01.49 |
| | Swiss | 29.85 | 02.92 | 17.02 | 02.56 | 18.76 | 03.02 |
| **Malay** | Standard | 47.98 | 43.61 | 46.81 | 35.27 | 27.86 | 22.86 |
| | Vernacular | 33.74 | 06.95 | 36.58 | 06.88 | 16.76 | 04.18 |
| | Moluccan | 17.11 | 02.04 | 12.03 | 06.11 | 09.03 | 05.89 |
| **Portuguese** | Brazilian | 57.96 | 57.09 | 55.54 | 52.98 | 42.62 | 35.96 |
| | European | 48.08 | 40.41 | 47.08 | 41.66 | 35.13 | 29.98 |
| **Swahili** | Coastal | 50.15 | 42.94 | 41.79 | 32.35 | 02.52 | 00.25 |
| | Congolese | 35.40 | 20.48 | 34.11 | 17.58 | 02.84 | 00.54 |

Table 4: Machine Translation Evaluations: BLEU Scores. Each model is evaluated across both directionalities with either the source or target specified as English. A higher BLEU score signifies better performance.

| Language | Dialect | Google NMT | | Meta NLLB | | Helsinki OpusMT | |
|---|---|---|---|---|---|---|---|
| | | di → EN | EN → di | di → EN | EN → di | di → EN | EN → di |
| **Arabic** | MSA | 0.851 | 0.890 | 0.826 | 0.872 | 0.590 | 0.834 |
| | Iraqi | 0.742 | 0.860 | 0.750 | 0.794 | 0.417 | 0.771 |
| | Omani | 0.792 | 0.805 | 0.801 | 0.841 | 0.502 | 0.811 |
| | Qatari | 0.743 | 0.814 | 0.742 | 0.787 | 0.416 | 0.768 |
| | Saudi | 0.792 | 0.806 | 0.788 | 0.800 | 0.447 | 0.777 |
| | Yemeni | 0.733 | 0.804 | 0.757 | 0.793 | 0.403 | 0.772 |
| | Jordanian | 0.777 | 0.785 | 0.780 | 0.789 | 0.416 | 0.767 |
| | Palestinian | 0.746 | 0.782 | 0.771 | 0.776 | 0.394 | 0.758 |
| | Lebanese | 0.681 | 0.797 | 0.747 | 0.765 | 0.352 | 0.745 |
| | Syrian | 0.764 | 0.788 | 0.776 | 0.777 | 0.382 | 0.755 |
| | Algerian | 0.661 | 0.780 | 0.704 | 0.777 | 0.378 | 0.752 |
| | Moroccan | 0.606 | 0.799 | 0.743 | 0.770 | 0.339 | 0.753 |
| | Libyan | 0.726 | 0.770 | 0.752 | 0.780 | 0.392 | 0.761 |
| | Tunisian | 0.521 | 0.787 | 0.685 | 0.757 | 0.329 | 0.747 |
| | Egyptian | 0.764 | 0.832 | 0.773 | 0.774 | 0.381 | 0.749 |
| | Sudanese | 0.792 | 0.805 | 0.782 | 0.819 | 0.449 | 0.796 |
| **Finnish** | Standard | 0.897 | 0.842 | 0.855 | 0.796 | 0.728 | 0.761 |
| | Kven | 0.777 | 0.754 | 0.581 | 0.642 | 0.692 | 0.701 |
| **Mandarin** | Mainland | 0.866 | 0.883 | 0.766 | 0.785 | 0.637 | 0.742 |
| | Taiwanese | 0.849 | 0.848 | 0.752 | 0.762 | 0.635 | 0.739 |
| **German** | Standard | 0.894 | 0.857 | 0.886 | 0.832 | 0.797 | 0.764 |
| | Low | 0.528 | 0.494 | 0.622 | 0.489 | 0.684 | 0.480 |
| | Swiss | 0.669 | 0.573 | 0.509 | 0.522 | 0.515 | 0.539 |
| **Malay** | Standard | 0.871 | 0.842 | 0.839 | 0.811 | 0.712 | 0.756 |
| | Vernacular | 0.819 | 0.652 | 0.805 | 0.665 | 0.654 | 0.627 |
| | Moluccan | 0.622 | 0.637 | 0.580 | 0.638 | 0.509 | 0.618 |
| **Portuguese** | Brazilian | 0.933 | 0.907 | 0.904 | 0.889 | 0.822 | 0.837 |
| | European | 0.909 | 0.858 | 0.889 | 0.859 | 0.809 | 0.820 |
| **Swahili** | Coastal | 0.855 | 0.857 | 0.823 | 0.818 | 0.247 | 0.199 |
| | Congolese | 0.728 | 0.749 | 0.717 | 0.732 | 0.257 | 0.200 |

Table 5: Machine Translation Evaluations: SentenceBERT Similarity. Each model is evaluated across both directionalities with either the source or target specified as English. A higher similarity score signifies better performance.

| Language | Dialect | Google USM | | OpenAI Whisper | | Meta XLS-R | |
|---|---|---|---|---|---|---|---|
| | | Spt | Cnv | Spt | Cnv | Spt | Cnv |
| **Spanish** | Argentinian | | 26.55% | | 33.76% | | 72.56% |
| | Chilean | | 25.80% | | 35.36% | | 74.95% |
| | Colombian | | 24.40% | | 34.32% | | 67.72% |
| | European | | 25.21% | | 30.75% | | 68.24% |
| | Mexican | | 24.55% | | 34.79% | | 68.78% |
| | Peruvian | | 27.52% | | 37.41% | | 71.93% |
| | Puerto Rican | | 36.88% | | 45.21% | | 84.83% |
| | Venezuelan | | 28.68% | | 35.69% | | 73.85% |
| **Arabic** | Iraqi | | 54.14% | | 134.09% | | 94.49% |
| | Omani | | 57.81% | | 157.52% | | 94.77% |
| | Saudi | | 50.51% | | 120.31% | | 93.12% |
| | Emirati | | 57.13% | | 154.92% | | 95.04% |
| | Jordanian | | 49.57% | | 129.44% | | 93.93% |
| | Lebanese | | 58.14% | | 136.57% | | 95.88% |
| | Palestinian | | 49.56% | | 121.88% | | 94.06% |
| | Syrian | | 48.78% | | 118.59% | | 94.59% |
| **Bengali** | Kamrupa | 60.98% | 51.11% | 207.58% | 190.83% | 103.31% | 101.16% |
| | Radha | 58.16% | 49.79% | 229.59% | 188.62% | 103.06% | 100.89% |
| | Varendra | 57.92% | 52.79% | 221.46% | 185.26% | 102.83% | 100.94% |
| **Georgian** | Eastern | 49.12% | 32.39% | 117.03% | 132.76% | 104.13% | 101.11% |
| | Western | 50.21% | 39.09% | 144.40% | 129.93% | 104.45% | 101.13% |
| **Tamil** | Central | 63.91% | 65.60% | 128.96% | 134.23% | 100.50% | 101.83% |
| | Madurai | 63.69% | 70.59% | 83.68% | 131.25% | 100.45% | 101.62% |
| | Northern | 65.12% | 68.07% | 150.17% | 131.10% | 100.01% | 101.60% |
| | Southern | 65.75% | 65.75% | 102.50% | 125.66% | 100.99% | 101.72% |
| | Western | 66.10% | 68.48% | 150.69% | 130.37% | 100.23% | 101.65% |
| **Telugu** | Central | 66.34% | 65.01% | 238.81% | 207.66% | 105.72% | 102.60% |
| | Eastern | 66.20% | 64.24% | 317.67% | 193.24% | 104.78% | 103.08% |
| | Northern | 65.37% | 67.24% | 320.80% | 211.71% | 105.57% | 103.27% |
| | Southern | 66.53% | 65.80% | 254.95% | 232.51% | 104.36% | 103.03% |
| **Tagalog** | Central | | | 110.49% | 90.21% | | |
| | Northern | | | 116.31% | 95.71% | | |
| | Southern | | | 94.09% | 105.98% | | |

| Language | Dialect | Google STT | | OpenAI Whisper + CV | | Meta XLS-R + CV | |
|---|---|---|---|---|---|---|---|
| | | Spt | Cnv | Spt | Cnv | Spt | Cnv |
| **Spanish** | Argentinian | | 55.36% | | 44.65% | | 62.58% |
| | Chilean | | 47.95% | | 37.03% | | 65.22% |
| | Colombian | | 49.71% | | 34.75% | | 59.61% |
| | European | | 45.22% | | 29.76% | | 57.27% |
| | Mexican | | 44.78% | | 43.14% | | 59.99% |
| | Peruvian | | 53.38% | | 42.68% | | 63.51% |
| | Puerto Rican | | 59.33% | | 51.63% | | 76.61% |
| | Venezuelan | | 53.47% | | 36.76% | | 65.82% |
| **Arabic** | Iraqi | | 72.47% | | 218.33% | | 95.39% |
| | Omani | | 73.93% | | 217.44% | | 96.16% |
| | Saudi | | 67.39% | | 185.53% | | 94.60% |
| | Emirati | | 74.39% | | 216.66% | | 95.93% |
| | Jordanian | | 72.91% | | 186.66% | | 94.94% |
| | Lebanese | | 82.27% | | 200.33% | | 96.37% |
| | Palestinian | | 73.78% | | 189.31% | | 95.15% |
| | Syrian | | 75.04% | | 195.73% | | 95.69% |
| **Bengali** | Kamrupa | 64.11% | 81.85% | 75.87% | 176.37% | 73.02% | 90.48% |
| | Radha | 63.09% | 79.85% | 68.35% | 144.77% | 68.34% | 88.84% |
| | Varendra | 65.82% | 80.37% | 74.17% | 153.00% | 71.54% | 89.59% |
| **Georgian** | Eastern | 64.64% | 81.44% | 143.95% | 222.75% | 68.61% | 88.56% |
| | Western | 66.95% | 81.71% | 165.49% | 210.67% | 72.83% | 88.11% |
| **Tamil** | Central | 64.97% | 86.39% | 73.25% | 108.17% | 77.71% | 98.46% |
| | Madurai | 63.82% | 86.54% | 73.00% | 109.13% | 76.79% | 98.16% |
| | Northern | 64.57% | 87.00% | 74.72% | 112.06% | 78.46% | 98.18% |
| | Southern | 65.13% | 86.75% | 79.97% | 110.75% | 81.36% | 98.25% |
| | Western | 64.13% | 84.39% | 77.67% | 106.23% | 77.07% | 97.58% |
| **Telugu** | Central | 63.94% | 84.35% | | | | |
| | Eastern | 63.36% | 84.91% | | | | |
| | Northern | 62.73% | 85.63% | | | | |
| | Southern | 64.62% | 85.87% | | | | |
| **Tagalog** | Central | 58.82% | 64.61% | | | | |
| | Northern | 58.16% | 65.01% | | | | |
| | Southern | 57.04% | 65.10% | | | | |

Table 6: Automatic Speech Recognition Evaluations: Word Error Rate. Each language is evaluated on three multilingual models and three monolingual models, with the IARPA Babel samples separated into Scripted (spt) and Conversational (cnv) classes. A lower word error rate signifies better performance.

| Language | Dialect | Google USM | | OpenAI Whisper | | Meta XLS-R | |
|---|---|---|---|---|---|---|---|
| | | Spt | Cnv | Spt | Cnv | Spt | Cnv |
| **Spanish** | Argentinian | | 17.54% | | 22.16% | | 35.73% |
| | Chilean | | 16.98% | | 23.10% | | 37.43% |
| | Colombian | | 16.61% | | 22.90% | | 32.17% |
| | European | | 18.27% | | 21.75% | | 32.94% |
| | Mexican | | 16.71% | | 23.40% | | 32.92% |
| | Peruvian | | 18.52% | | 25.03% | | 35.87% |
| | Puerto Rican | | 25.75% | | 31.33% | | 46.90% |
| | Venezuelan | | 19.80% | | 24.22% | | 37.01% |
| **Arabic** | Iraqi | | 23.97% | | 104.15% | | 53.48% |
| | Omani | | 27.56% | | 128.51% | | 55.32% |
| | Saudi | | 21.94% | | 91.25% | | 52.54% |
| | Emirati | | 26.45% | | 121.32% | | 55.08% |
| | Jordanian | | 20.57% | | 95.77% | | 52.01% |
| | Lebanese | | 27.38% | | 99.87% | | 54.49% |
| | Palestinian | | 20.06% | | 99.87% | | 52.53% |
| | Syrian | | 19.83% | | 82.85% | | 52.02% |
| **Bengali** | Kamrupa | 41.02% | 29.56% | 231.39% | 218.35% | 93.63% | 94.60% |
| | Radha | 36.26% | 30.91% | 268.22% | 213.89% | 93.34% | 94.43% |
| | Varendra | 37.27% | 31.28% | 220.76% | 207.11% | 93.30% | 94.25% |
| **Georgian** | Eastern | 35.15% | 13.83% | 336.56% | 298.09% | 72.64% | 88.39% |
| | Western | 36.11% | 18.16% | 349.79% | 291.62% | 74.14% | 86.77% |
| **Tamil** | Central | 42.42% | 29.44% | 63.03% | 100.79% | 59.90% | 93.04% |
| | Madurai | 40.91% | 38.91% | 49.65% | 90.71% | 55.39% | 92.53% |
| | Northern | 41.04% | 34.72% | 78.49% | 86.91% | 58.23% | 92.47% |
| | Southern | 43.58% | 31.01% | 69.37% | 86.06% | 59.82% | 92.35% |
| | Western | 43.42% | 37.36% | 107.68% | 88.58% | 56.77% | 92.13% |
| **Telugu** | Central | 39.95% | 35.86% | 251.07% | 275.66% | 97.61% | 95.46% |
| | Eastern | 41.20% | 33.14% | 307.03% | 271.42% | 96.86% | 95.46% |
| | Northern | 40.95% | 37.10% | 233.57% | 279.64% | 96.89% | 95.89% |
| | Southern | 42.03% | 36.01% | 275.70% | 282.87% | 96.70% | 95.77% |
| **Tagalog** | Central | | | 81.86% | 62.82% | | |
| | Northern | | | 87.30% | 66.62% | | |
| | Southern | | | 66.92% | 69.32% | | |

| Language | Dialect | Google STT | | OpenAI Whisper + CV | | Meta XLS-R + CV | |
|---|---|---|---|---|---|---|---|
| | | Spt | Cnv | Spt | Cnv | Spt | Cnv |
| **Spanish** | Argentinian | | 43.08% | | 27.71% | | 29.56% |
| | Chilean | | 33.53% | | 22.74% | | 30.89% |
| | Colombia | | 38.25% | | 22.01% | | 27.93% |
| | European | | 35.47% | | 18.89% | | 27.31% |
| | Mexican | | 31.45% | | 26.99% | | 28.30% |
| | Peru | | 40.95% | | 26.77% | | 30.30% |
| | Puerto Rican | | 44.26% | | 32.40% | | 40.68% |
| | Venezuelan | | 40.83% | | 22.96% | | 31.97% |
| **Arabic** | Iraqi | | 44.05% | | 191.00% | | 51.91% |
| | Omani | | 49.02% | | 189.29% | | 54.16% |
| | Saudi | | 40.39% | | 152.65% | | 51.88% |
| | Emirati | | 48.28% | | 188.34% | | 54.28% |
| | Jordanian | | 45.74% | | 157.30% | | 51.19% |
| | Lebanese | | 55.58% | | 170.02% | | 53.70% |
| | Palestinian | | 47.21% | | 158.14% | | 51.35% |
| | Syrian | | 49.19% | | 168.76% | | 51.45% |
| **Bengali** | Kamrupa | 41.55% | 69.20% | 39.32% | 139.69% | 30.26% | 51.65% |
| | Radha | 39.86% | 67.09% | 31.26% | 108.85% | 27.39% | 49.09% |
| | Varendra | 42.39% | 68.01% | 37.11% | 117.18% | 29.86% | 49.92% |
| **Georgian** | Eastern | 44.43% | 53.26% | 100.37% | 183.55% | 18.16% | 35.25% |
| | Western | 46.48% | 52.86% | 110.34% | 176.89% | 20.15% | 34.67% |
| **Tamil** | Central | 41.54% | 65.28% | 32.56% | 70.98% | 30.02% | 59.84% |
| | Madurai | 40.35% | 65.60% | 33.71% | 73.86% | 26.98% | 59.70% |
| | Northern | 40.94% | 66.49% | 33.79% | 75.13% | 29.69% | 59.71% |
| | Southern | 41.37% | 65.62% | 40.76% | 72.77% | 32.30% | 59.17% |
| | Western | 40.59% | 62.21% | 34.51% | 69.72% | 28.16% | 57.29% |
| **Telugu** | Central | 40.76% | 69.31% | | | | |
| | Eastern | 40.58% | 69.53% | | | | |
| | Northern | 39.09% | 71.41% | | | | |
| | Southern | 42.03% | 71.33% | | | | |
| **Tagalog** | Central | 44.89% | 44.39% | | | | |
| | Northern | 44.69% | 44.85% | | | | |
| | Southern | 43.85% | 44.95% | | | | |

Table 7: Automatic Speech Recognition Evaluations: Character Error Rate. Each language is evaluated on three multilingual models and three monolingual models, with the IARPA Babel samples separated into Scripted (spt) and Conversational (cnv) classes. A lower character error rate signifies better performance.