# OpenReview forum: "Quantifying the Dialect Gap and its Correlates Across Languages"
_EMNLP/2023/Conference — EMNLP 2023 Findings_

### Official Review · Reviewer_J4pE · 2023-07-31

**Soundness:** 3

**Excitement:**

2: Mediocre: This paper makes marginal contributions (vs non-contemporaneous work), so I would rather not see it in the conference.

**Missing References:**

The work would benefit from a much more significant literature review of existing NLP methods for dialects and related languages. VarDial has many relevant works, and prior methods have studied the challenges of translating to and from dialects. Contextualizing the work in prior work will help highlight areas where the analysis can be further expanded to add on to existing literature, rather than reiterate it.

Relevant References to Section 2:
Surafel Melaku Lakew, Aliia Erofeeva, and Marcello Federico. 2018. Neural Machine Translation into Language Varieties. In Proceedings of the Third Conference on Machine Translation: Research Papers, pages 156–164, Brussels, Belgium. Association for Computational Linguistics.

Iuliia Nigmatulina, Tannon Kew, and Tanja Samardzic. 2020. ASR for Non-standardised Languages with Dialectal Variation: the case of Swiss German. In Proceedings of the 7th Workshop on NLP for Similar Languages, Varieties and Dialects, pages 15–24, Barcelona, Spain (Online). International Committee on Computational Linguistics (ICCL).

Zampieri, M., Nakov, P., & Scherrer, Y. (2020). Natural language processing for similar languages, varieties, and dialects: A survey. Natural Language Engineering, 26(6), 595-612.

Sun, J., Sellam, T., Clark, E., Vu, T., Dozat, T., Garrette, D., ... & Gehrmann, S. (2022). Dialect-robust evaluation of generated text. arXiv preprint arXiv:2211.00922.

Salloum, W., & Habash, N. (2012, December). Elissa: A dialectal to standard Arabic machine translation system. In Proceedings of COLING 2012: Demonstration Papers (pp. 385-392).

Demszky, D., Sharma, D., Clark, J. H., Prabhakaran, V., & Eisenstein, J. (2021, June). Learning to Recognize Dialect Features. In Proceedings of the 2021 Conference of the North American Chapter of the Association for Computational Linguistics: Human Language Technologies (pp. 2315-2338).

Caleb Ziems, William Held, Jingfeng Yang, Jwala Dhamala, Rahul Gupta, and Diyi Yang. 2023. Multi-VALUE: A Framework for Cross-Dialectal English NLP. In Proceedings of the 61st Annual Meeting of the Association for Computational Linguistics (Volume 1: Long Papers), pages 744–768, Toronto, Canada. Association for Computational Linguistics.

Deas, N., Grieser, J., Kleiner, S., Patton, D., Turcan, E., & McKeown, K. (2023). Evaluation of African American Language Bias in Natural Language Generation. arXiv preprint arXiv:2305.14291.

Eisenstein, J., Prabhakaran, V., Rivera, C., Demszky, D., & Sharma, D. (2023). MD3: The Multi-Dialect Dataset of Dialogues. arXiv preprint arXiv:2305.11355.



**Paper Topic And Main Contributions:**

This paper looks at the performance disparity for dialects inside of modern NLP technology. They quantify this disparity for large-scale industrial Machine Translation and ASR systems between several groups of dialects and related languages using existing benchmarks which cover dialects and related languages in 7 languages for MT and 7 other languages for ASR. They then study the correlations between these performance results and high level economic indicators, population size, and metrics of linguistic similarity. For all languages, variation is shown between languages. Additionally, a correlation is shown between performance for a dialect and it's lexical and phonetic similarity with high-resource dialects.

**Questions For The Authors:**

A) Line 325 mentions a manual analysis of a random sample of data. Who did the manual analysis and what was their background? How was the data sample to make sure that it was representative of the population, rather than some speakers? Are there existing linguistic resources that this analysis agrees with or is this an entirely novel contribution of the work?

B) For the correlations, how well fit are your regressions on each variable? It's hard to judge the meaning of any of these without metrics (at least in the appendix) for goodness of fit.

C) Many of the economic and population metrics used seem like they likely correlate with dataset size. Did you control for dataset size in this analysis? If not, how likely is it that these trends are just a function of dataset size?

**Reasons To Accept:**

- The work looks at discrepancies across both Multilingual and Multidialectal NLP simultaneously. Looking at these dimensions simultaneously is important to understand the effects of language variation at multiple scales and multiple levels of resourcedness. By doing so, the authors get larger samples which could serve as a more effective foundation for causal analysis of what drives the dialect gap.

- The similarity analyses are interesting as a predictor of transfer performance between dialects. Understanding how the similarity metrics the authors use could be used to identify training data from other dialects which might lead to positive transfer could be an interesting pathway to addressing the data size issues the authors discuss later in the work.

**Reasons To Reject:**

- The paper does not significantly extend the quantification of the dialect gap beyond existing published work. Several works have quantified the effects of dialect on machine translation systems for different languages. Similarly, several works have quantified the effects of dialect on ASR for different languages. This paper does not introduce any resources that did not exist in prior work (see Appendix A, all resources existed prior to this work). While evaluation of more recent systems is informative, the evaluation does not particularly shift the understanding of dialect disparities in NLP nor offer novel pathways to resolving the gap.

- The section dedicated to causal analysis of the dialectal gap requires a more formalized statistical analysis plan to justify the use of "causes across languages" in the title. Only the correlations between performance and different features are evaluated, with many of these features being certainly not directly causal in their relationship with performance. Even at just the level of correlations, some results seem questionable (GDP per Capita and HDI which are themselves colinear show sometimes contradictory trends). I think the work would be stronger if it looked at a smaller set of factors in a more thorough manner, rather than covering a broad range of factors shallowly.

- The paper does not seem particularly well-suited to the theme. None of the models evaluated appear to be LLMs as the term is used to refer to general purpose multi-task pretrained models such as LLaMA, GPT series, FLAN, FALCON, Pythia, etc. Instead, each one is a purpose trained model for a single task (either Machine Translation or Automatic Speech Recognition). The titling suggests that the paper will quantify the dialect gap in this broad general purpose setting, but only covers Machine Translation and ASR which are not new challenges brought with LLMs. The work would be a stronger fit in the Multilinguality track (Subtrack for Dialects and Related Languages).

**Reproducibility:**

5: Could easily reproduce the results.

**Reviewer Confidence:**

5: Positive that my evaluation is correct. I read the paper very carefully and I am very familiar with related work.

**Typos Grammar Style And Presentation Improvements:**

Figure 3(a): It's not explained how these projections are fit to the data and to what degree we should trust them, especially given grokking phenomena common in deep learning models. Explain how they were fit and how they were tested!

Figure 2: It's very difficult to actually extract the correlations from these plots - not to mention they are space inefficient. I think a table or a bar chart of correlations would be more effective ways to show this data as is common for regressions in causal inference.

Somtimes (like table 1) the model is referred to as XLS-R, other times (like figure 3b) it's referred to as XSL-R. Make this consistently the correct XLS-R!

---

> ### Author Rebuttal · Authors · 2023-08-28
>
> Thank you for taking the time to read our paper and leaving such detailed feedback. We are glad you found our work to be interesting in its multidimensional approach and agree that the linguistic metrics we identify as strong correlates are valuable as starting points for improving dialect performance in future work. We address your comments and concerns below:
>
> > Does this paper significantly extend the quantification of the dialect gap beyond prior work?
>
> While we build on prior work, we believe that our work differs in several key ways from prior work studying dialects in NLP. This work is unique in its scope - prior work on dialects in NLP largely focuses on a single language or language family. Here, we are instead looking at broader trends in performance for two high-impact tasks: machine translation and automatic speech recognition. **By examining performance across dialects of both low- and high-resource languages, we can extract trends to identify routes for future, language-agnostic work in dialect NLP.** Thank you for the additional references, including other analyses of dialectal performance (Lakew et al., 2018; Nigmatulina et al., 2020; Zamipieri et al., 2020; Sun et al., 2022), methods of improving dialectal performance (Salloum & Habash, 2012; Demszky et al., 2021) and some other related works published at or after the EMNLP deadline (Ziems et al., 2023; Deas et al., 2023; Eisenstein et al., 2023). We will discuss additional related works in the final version of our paper.
>
> > Are these models LLMs?
>
> There is no single definition of “large language models,” but most of these models (e.g. Whisper) satisfy proposed requirements such as being trained on massive amounts of scraped internet data, having extremely large parameter sizes, and having the ability to apply to multiple tasks without further training . We had a hard time ourselves deciding on the single ‘best-aligned’ track for the work, which is why we indeed submitted this paper to both the LLM track and the multilinguality track: it straddles the line between both, and we agree that it can be argued to belong in either track.
>
> > How was the phonetic analysis (line 325) conducted and is this novel work?
>
> The formant analysis done in this paper is grounded in linguistic theory as discussed in various linguistic dialectology papers, such as (Ghorshi et al., 2008). This citation will be included in the final version. The analysis was conducted by one of the authors who is a linguist and was decided upon after consulting a phoneticist to validate the methodology. The samples were taken randomly across participants to ensure that no one participant’s speech patterns would significantly impact the results (i.e. when there were greater than 10 participants, each sample was from a unique participant, otherwise equal number of samples (+ or - 1) were taken from each participant).
>
> > How well fit are the correlation regressions?
>
> Thank you for raising this point. The Pearson correlation coefficients for the correlations are as follows (where the score axis is inverted for WER (ASR), so negative coefficients indicate a positive correlation in performance):
>
> | | MT →en | MT →di | ASR mult | ASR mono |
> | ---- | ---- | ---- | ---- | ---- |
> | GDP | 0.11 | 0.16 | -0.25 | -0.13 |
> | GPC | 0.16 | 0.31* | 0.44* | 0.16 |
> | POP | 0.04 | -0.13 | -0.30 | -0.00 |
> | HDI | -0.06 | -0.03 | 0.23 | 0.09 |
> | LEX | 0.48* | 0.69* | -0.70* | -0.57* |
> | PHO | - | - | -0.49* | -0.63* |
> *p<0.05
>
> This demonstrates that **only the correlation between lexical and phonological features and performance is statistically significant**. Thank you for pointing out that these should be included in the paper - the values themselves will be reported in Section 5 with the p-values being included in the Appendix to strengthen the argument. Additionally, we plan to change the title from “...Causes Across Languages” to “...Correlates Across Languages” to better reflect the content of the analysis.
>
> > Are these trends just a factor of dataset size (which might be correlated with socioeconomic factors)?
>
> No, since we kept dataset sizes uniform across dialects of each language and pre-training was done dialect-agnostically. We are primarily interested in performance relative to the standard dialect rather than absolute performance, so the correlates are taken with regards to % of the standard score. Certain languages and dialects may have more data available at pretraining due to socioeconomic factors, but **our analysis shows that any correlation between these factors and performance is shallow at best and contradictory at worst, as you mentioned**. The only consistently statistically significant correlates are lexical and phonetic similarity as shown by the correlation coefficients calculated above. These trends might be a result of dataset makeup, as discussed in Section 6.2, but that is part of the point of this paper - pretraining datasets need to be better weighted across dialects if we don’t want performance to disproportionately be weighed in benefit of the standard forms.
>
> > Improve Figures 2 and 3.
> Thank you for pointing out this important issue, we will fix this in the final version. We'll structure Figure 2 with the information provided above to be more explicitly clear and make it larger with the additional space available in the final version. For Figure 3, the projection lines mostly serve to emphasize the different trends seen across languages and directionalities; following your feedback, we will remove this in the final version.

---

### Official Review · Reviewer_7UB3 · 2023-08-04

**Soundness:** 4

**Excitement:**

4: Strong: This paper deepens the understanding of some phenomenon or lowers the barriers to an existing research direction.

**Paper Topic And Main Contributions:**

This paper presents a study on the performance of LLMs in MT and ASR for diverse dialects, usually a "major/standard" and one or more "minor" dialects. The performance is correlated with various socio-economic and linguistic metrics to better understand what could be of influence on the performance on minority dialects of the tested LLMs. This leads to sometimes surprising and counterintuitive results - which is an interesting starting point for data development for dialectal NLP. A clear conclusion is that the more lexically or phonetically similar a minority dialect to the standard the better the performance of the tested models.

**Reasons To Accept:**

The paper is very comprehensive in the analysis of the two chosen tasks, the analysis and explanation of trends as well as the directions for future research. Also, it addresses the important topic of linguistic variety in NLP, not just at the language level, but at the level of dialects. The authors choice of socio-economic and linguistics metrics was thoughtful and a good illustration of the fact that language is less homogeneous than some assume.

**Reasons To Reject:**

I do not currently see any reasons why this paper should be rejected.

**Reproducibility:**

4: Could mostly reproduce the results, but there may be some variation because of sample variance or minor variations in their interpretation of the protocol or method.

**Reviewer Confidence:**

4: Quite sure. I tried to check the important points carefully. It's unlikely, though conceivable, that I missed something that should affect my ratings.

**Typos Grammar Style And Presentation Improvements:**

line 008: have been conducted -> have been mainly conducted?

line 024: one-size approach -> one-size-fits-all approach

line 186: socioeconomic -> socio-economic

line 246: BLEU score -> reference?

line 578: makes out system -> makes our system?

line 1107: Appen PTY Ltd, Sydney and Australia -> correct reference to not include Sydney and Australia

line 1153ff: by splitting by taking -> by splitting

References: Please check capitalisation in references (e.g.nlp, bert) and correct where necessary.

---

> ### Author Rebuttal · Authors · 2023-08-28
>
> Thank you for your review and for your kind words. We are glad that you found our paper comprehensive and thoughtful in its approach. We hope that, as you said, this work will be used as a starting point for data collection and development at the dialectal level in future work. Your grammar and spelling corrections are helpful and will be incorporated into the final version.

---

### Official Review · Reviewer_V96V · 2023-08-11

**Soundness:** 4

**Excitement:**

4: Strong: This paper deepens the understanding of some phenomenon or lowers the barriers to an existing research direction.

**Missing References:**

- This paper shows how transfer performance can be improved by using data/models of closely-related (i.e. linguistically similar) high-resource languages, exploiting the connection between linguistic similarity and transferability of model performance: https://aclanthology.org/2023.nodalida-1.74.pdf
- This paper looks at correlations between language data resources, inclusion in pretrained multilingual models, and various social factors of speaker bases (e.g. population, GDP): https://arxiv.org/pdf/2210.08523.pdf Although they don’t examine model performance, it seems relevant as it may offer insights into relationships between some of the dependent variables in your paper.

**Paper Topic And Main Contributions:**

This paper presents an evaluation of 6 LLMs from Google, Meta, and OpenAI on the tasks of machine translation (MT) and automatic speech recognition (ASR), for at least 30 language varieties across 7 languages per task. The authors find that disparities in performance correlate more with linguistic proximity (i.e. lexical and/or phonetic similarity) to well-resourced language varieties than with GDP, population, or HDI of the speaker base. They also find that training data size and construction procedure is a significant factor but cannot entirely explain performance disparities. Finally, they discuss how to address this gap in performance.

**Questions For The Authors:**

a) Section 4 mentions other ways of capturing linguistic similarity via syntactic, lexical, or morphological features. What was the reason for choosing the lexical and phonetic similarity metrics over one of these feature-based approaches?

**Reasons To Accept:**

- The task is an important one which is relevant to current issues in NLP.
- The findings are interesting and have important implications.
- The study examines an impressive range of language varieties across language families and across high- and low-resource languages, and examines correlations with social as well as linguistic factors.
- Experimental design choices (e.g. models, languages, metrics) are well argued for.
- The paper is clearly written and well organized, with detailed appendices.

**Reasons To Reject:**

- The title is misleading; this paper examines correlations, not causes.
- The multiple datasets include data from different sources and domains (e.g. religious, media, Wikipedia), and likely have different annotation/transcription practices. This limits the comparability across all the datasets and languages.
- The lexical similarity metric seems like it may also be capturing orthographic similarity between language varieties, which is something quite different from linguistic similarity.


**Reproducibility:**

4: Could mostly reproduce the results, but there may be some variation because of sample variance or minor variations in their interpretation of the protocol or method.

**Reviewer Confidence:**

4: Quite sure. I tried to check the important points carefully. It's unlikely, though conceivable, that I missed something that should affect my ratings.

**Typos Grammar Style And Presentation Improvements:**

- I was not able to understand Figure 1a, or what it was meant to tell us; please consider elaborating on the explanation in Section 4 or in the caption.

---

> ### Author Rebuttal · Authors · 2023-08-28
>
> Thank you for taking the time to read our paper. We are glad you found our paper interesting, well-organized, and well-argued, with an impressive range of language varieties. We have addressed your comments for improvement below:
>
> > Title should reference “correlates” rather than “causes”.
>
> We agree, and the title will be updated to reflect that in the final version.
>
> > Are the datasets comparable due to differences in domains and collection practices?
>
> Yes, in the way we use them. Within each language, only one dataset was used across dialects, so the comparisons between the performance of dialects within languages are sound. All the performance comparisons in this paper are done *within languages*, with percentage differences between dialects of the same language being used in Figure 2, so even if a specific dataset is “easier,” it would not have an impact on the overall analysis. In general, we are focusing on the relative performance between dialects of each language and comparing these *relative performances* across languages because achieving a controlled and uniform environment when dealing with pre-trained models for multiple languages is difficult.
>
> > Why were lexical and phonetic features used over the feature-based metrics discussed in Section 4?
>
> The syntactic, morphological, and lexical feature metrics discussed in Section 4 have been used when quantifying the linguistic similarity between languages, but these features do not necessarily apply the same way or to the same extent when examining dialects. Dialects are (usually) linguistically closer together than languages, and as such broader trends such as word order, affixation, etc. will be the same across dialects. Dialects primarily arise due to lexical and phonetic shifts, with syntactic and morphological shifts being less common and usually appearing after a more extended period of divergence. To keep our analysis applicable to all the dialects examined, these two metrics were prioritized.
>
> > Is the lexical variation metric capturing orthographic variation too?
>
> Yes, but that’s not a bad thing. Analyzing lexical variation impacts MT and ASR differently, so we will address them individually. In MT, orthographic variations can result from two different sources: lexical shift (where new words or slang are introduced) and phonetic shift (where spelling shifts to match phonetic drift). Both of these are common in dialects, and if MT systems are to work on dialectal text, they will need to function regardless of why the lexicon has changed. So, lexical similarity is a good metric to use even if it captures variation from both of these sources. For ASR, lexical variation usually does not reflect orthographic variation because the annotators of these datasets were instructed to transcribe into the standard form unless the word is unique to the dialect. Here, the lexical similarity metric purely captures lexicon drift so we also utilize our phonetic similarity metric to account for phonetic changes.
>
> > Figure 1a is difficult to understand.
>
> Thank you for pointing out this issue. To resolve this problem, the figure will be made larger and more space-efficient in the final version. The figure itself is a violin plot, where each bar is composed of calculations of Spearman’s Rank Correlation Coefficient across 100 random splits of the data as calculated in Appendix B. A higher coefficient translates to high levels of similarity in the lexicon of the two dialects, and a lower variance translates to a lower likelihood that the similarity scores are due to poor sampling. In the final version, more information about the content of the figure will be explained in Section 4 rather than just referencing Dunn (2021), which is where the structure of the figure is pulled from.
>
> > Missing references.
>
> Thank you for bringing these two papers to our attention. The first paper (Snæbjarnarson et al., 2023) will be referenced in the future work section as a way to apply what has been established by this paper to improve performance. The second paper (Ranathunga & Silva, 2022) will be referenced in our related works as evidence of unbalanced training sets and may additionally be brought up in Section 6.

---

### Official Review · Reviewer_rTFa · 2023-08-12

**Typos Grammar Style And Presentation Improvements:** It was very well written.
**Soundness:** 3

**Excitement:**

3: Ambivalent: It has merits (e.g., it reports state-of-the-art results, the idea is nice), but there are key weaknesses (e.g., it describes incremental work), and it can significantly benefit from another round of revision. However, I won't object to accepting it if my co-reviewers champion it.

**Paper Topic And Main Contributions:**

The paper focuses on the challenges related to linguistic diversity and dialectal variations in NLP, particularly in the context of Large Language Models (LLMs). It explores the lack of attention to under-resourced languages and dialects, as well as the complexities in recognizing different speech forms.

Main Contributions:
1. Highlighting Linguistic Diversity Issues: Emphasizes gaps in research, especially outside Europe and East Asia.
2. Exploration of Dialectal Models: Discusses the capabilities and limitations of LLMs in handling under-resourced languages and dialects.
3.Speech Recognition Analysis: Investigates how training data affects the performance of speech recognition models across dialects.

In essence, the paper offers insights and analysis on linguistic disparities in NLP and suggests directions for developing more inclusive and effective models across languages and speech forms.

**Questions For The Authors:**

A. Could you elaborate on the unexpected correlations and negative results found in the study? How do these findings fit into the broader context of the field, and what additional analysis might be required to interpret them fully?

B. Expand on Linguistic Analysis of Dialects?

C.  Why was there no qualitative analysis presented?

**Reasons To Accept:**

Strengths:

- The paper provides an extensive examination of linguistic diversity in NLP, highlighting key disparities and areas that need more attention.
- Paper is well-written, easy to follow and well motivated.
- The paper's exploration of dialectal differences and under-resourced languages fills a significant gap in the current literature, adding depth to the understanding of language and dialect support in NLP systems.
The chosen tasks (MT and ASR) are well suited for the analysis, facilitating a detailed and relevant examination of the subject matter.

**Reasons To Reject:**

The unexpected correlations mentioned in the text (such as negative correlation with wealth in ASR or inconsistent findings between MT and ASR) are acknowledged but not deeply analyzed or explained. This leaves gaps in understanding and may limit the applicability of the findings.
The presentation of the results, including the graphs, may not be clearly laid out or properly explained, hindering interpretation, especially for readers coming from different fields.
There is a noticeable absence of qualitative analysis in the paper.
The linguistic analysis of dialects is not properly explained, which can make it difficult for readers to fully grasp this essential aspect of the research.

The lack of rigor in some experiments, gaps in analysis, unclear presentation of results, and unexplored negative findings may limit the paper's contribution to the field and its applicability to real-world scenarios.

**Reproducibility:**

5: Could easily reproduce the results.

**Reviewer Confidence:**

2: Willing to defend my evaluation, but it is fairly likely that I missed some details, didn't understand some central points, or can't be sure about the novelty of the work.

---

> ### Author Rebuttal · Authors · 2023-08-28
>
> Thanks to the reviewer for reading our paper. We are glad to hear that you find this work to be a detailed and relevant examination that highlights important disparities, especially for languages from under-served parts of the world. We are encouraged that you find our work to be well-written, easy to follow, and well-motivated. Please see our response below:
>
> > How do the unexpected correlations/negative results fit into the broader context?
>
> Previous work has shown that disparities between languages in NLP technology are loosely correlated with socioeconomic factors such as GDP and population size (Blasi et al., 2021). As such, it would be logical to assume that dialects would follow a similar trend, with more widely spoken or wealthier dialects being better supported. However, this is not the trend we discovered; instead, there is *no significant correlation* between the performance of models on dialects and any socio-economic factors. The only correlations were found to be with linguistic proximity to the “standard” dialect. This demonstrates that **assumptions established in the study of low-resource languages do not necessarily translate to dialects**. Additionally, this means that linguistically informed similarity analysis will be extremely valuable in creating methods for shrinking the dialect performance gap. We will clarify this in the next version of the paper.
>
> > Include some qualitative analysis.
>
> This is a great idea and inspired us to run an error analysis on languages discussed in Figure 3 to identify whether the presence of hand-selected “dialectal” words seems to lead to lower performance. Examples of sentences with similar English translations but different levels of accuracy across dialects will be included in the final version, with their lexical variations being compared to existing literature on dialectal lexicon.  An example:
> Both of these sentences are officially translated as “Children need loving.”
> Standard German - Kinder brauchen Liebe.
> NLLB - Children need love.
> Swiss German - Chind bruchet Liebi.
> NLLB - Chind broke my heart.
> In this example, “chind” is a Swiss German word for child and is not picked up by the model, which assumes it is a proper noun. Additionally, the conjugation of both of the other words is spelled differently, which leads to a completely incorrect interpretation of the sentence. Although there is evidence here that German has variation outside of lexical and phonetic differences, calculating the level of lexical divergence alone is enough to predict the degree to which this model will struggle with the dialect.
>
> > Expand on the linguistic analysis of dialects.
>
> Further information on the process of linguistic analysis is already included in Appendix B, but this section will be clarified with more details on the similarity computation processes and more background on how these techniques were selected from past work in computational linguistics.
>
> > The figures are difficult to read.
>
> We agree and with the additional page in the final version of the paper, we will redo the figures to be easier to read and bigger so that the labels are easier to parse. With the addition of more details concerning our linguistic analysis, there will be enough clarification to assist even those without specific background knowledge in linguistics or computer science to understand the analysis.
>
> > There is a lack of rigor and gaps in our analysis.
>
> We confidently establish in this paper that dialects perform differently from low-resource languages and that linguistic similarity is a valuable starting point for future work. We study the phenomenon of the dialect performance gap across 2 tasks (with 2 variations each), 3 models per task, and 7 languages per task. We find consistent trends across these 84 settings, including a correlation between linguistic proximity to the standard dialect and better performance of a dialect. We also find *no significant correlation* between factors such as wealth, population size, and level of development and the corresponding dialect’s performance in the cases studied. While the analysis of such complex sociological and linguistic phenomenon could fill a much larger work, we are confident in our findings. Our findings are further supported by Section 6, where we use the idea of addressing the linguistic diversity of training data through finetuning as a starting point to address the dialect performance gap.

---

### Meta-Review · Area_Chair_G46X · 2023-09-25

**Recommendation:** 3

**Metareview:**

This paper evaluates the performance of large language models (LLMs) on machine translation (MT) and automatic speech recognition (ASR) for different languages and dialects. The paper uses six LLMs from Google, Meta, and OpenAI, and tests them on 30 language varieties across 7 languages per task.. The paper measures the performance disparity between major/standard and minor dialects, and analyzes the factors that influence it. The paper finds that linguistic similarity to well-resourced language varieties is more important than socio-economic indicators or training data size. The paper also discusses the challenges and opportunities for improving dialectal NLP and hence has merits for promoting research for dialects.

---

### Decision · Program_Chairs · 2023-10-07

**Decision:**

Accept-Findings

**Comment:**

This paper evaluates the performance of large language models (LLMs) on machine translation (MT) and automatic speech recognition (ASR) for different languages and dialects. The paper uses six LLMs from Google, Meta, and OpenAI, and tests them on 30 language varieties across 7 languages per task.. The paper measures the performance disparity between major/standard and minor dialects, and analyzes the factors that influence it. The paper finds that linguistic similarity to well-resourced language varieties is more important than socio-economic indicators or training data size. The paper also discusses the challenges and opportunities for improving dialectal NLP and hence has merits for promoting research for dialects.